# New Experimental Data on Partial Pressures of Gas Phase Components over Uranium-Zirconium Carbonitrides at High Temperatures and Its Comparative Analysis

G. S. Bulatov and Konstantin E. German *

Frumkin Institute of Physical Chemistry and Electrochemistry of Russian Academy of Sciences,
119071 Moscow, Russia
* Correspondence: guerman_k@mail.ru

**Abstract:** Data from the literature were analyzed and experimental data were obtained on the sublimation of uranium-zirconium carbonitrides with different contents of carbon, nitrogen and oxygen impurities in a wide temperature range (1773–2323 K). The composition of the gas phase was determined and the analytical dependences of the partial pressures of its components on temperature were calculated. It was shown that the sublimation of uranium-zirconium carbonitrides occurs incongruently with the predominant loss of nitrogen, which led to a shift in their compositions to the side richer in carbon. The chemical mechanism of sublimation was considered, according to which oxygen impurities in these materials contribute to the additional release of nitrogen and the appearance of oxide components $UO$, $UO_2$ and $CO$ in the gas phase. The introduction of zirconium carbonitrides and an increase in the carbon content led to a decrease in the partial pressures of uranium monoxide and nitrogen, thereby increasing the thermal stability of materials.

**Keywords:** high-temperature nuclear fuel; uranium-zirconium carbonitrides; mass spectrometry; gas phase; partial pressures; oxygen; temperature

## 1. Introduction

Fissile materials based on refractory compounds of uranium, plutonium or thorium correspond most fully to the operating conditions in high-temperature nuclear reactors (with gas or liquid-metal coolants, as well as thermionic converters—TEC). These compounds include: dioxides, mono and dicarbides, mononitrides, carbonitrides, as well as sulfides and phosphides of fissile metals.

To use these compounds in nuclear reactors, it is necessary to know the characteristics of materials: thermophysical (melting point, vapor pressure, thermal expansion, thermal conductivity), chemical (compatibility with cladding materials, oxidizing media), mechanical (strength, creep), nuclear physics (resistance to neutron irradiation, swelling, etc.).

Uranium carbonitrides—UCN and uranium-zirconium carbonitrides—(UZrCN) with a zirconium content of ~3–4% wt. were currently considered as advanced nuclear fuel for use in high-temperature gas-cooled nuclear reactors [1–3]. The thermodynamic properties, including the partial pressures of the components over UCN, have been investigated in a number of works [4–9]. Most beneficial items of the application of this fissile material have been considered and summarized in the reviewing book by Alexeev S.V., and Zaitsev V.A. [1].

The study of the properties of refractory compounds of fissile metals (mainly uranium) in a wide temperature range is the subject of a significant number of works, the results of which have been summarized in a number of monographs [10–13]. However, due to the fact that the properties of nuclear fuel depend on many factors (production technology, processing conditions, the presence of impurities, defects, etc.), the data given in the

literature often cannot be used when considering specific fuel compositions, which requires additional research.

The high-temperature properties of uranium dioxide, a fuel currently used in various types of reactors, have been studied in much detail. Less studied among those used in practice were carbide fuels—uranium mono and dicarbide.

Fuels based on uranium mononitride, as well as carbonitrides, although they were recognized as promising for use in certain types of nuclear power plants, for example, fast neutron reactors [11], have not yet found wide application due to the limited experimental data on their properties, and also because of the relative complexity of the industrial technology for their production.

Compounds based on uranium mononitride or carbide were in interstitial phases in which the metalloid atom (nitrogen, carbon) was located in the metal lattice at interstitial positions. Interatomic bonds in uranium nitrides and carbides were mostly metallic, which led to electronic conductivity, the presence of metallic luster. At the same time, the materials retained the high-strength, brittleness and hardness characteristics of ceramics.

Uranium nitrides and carbides have two main advantages over dioxide: firstly, the density of metal atoms in them is higher: for UN it is $13.52.10^{-3}$ kg·m$^{-3}$, for UC—$12.97.10^{-3}$ kg·m$^{-3}$, for $UC_2$—$11.68.10^{-3}$ kg·m$^{-3}$, and for $UO_2$—$9.65.10^{-3}$ kg·m$^{-3}$ as it has been shown in [10] and, secondly, they have a significantly higher thermal conductivity: for UN it is equal to 20 W m$^{-1}$ K$^{-1}$ (at 1023 K), for UC—21.6 W m$^{-1}$ K$^{-1}$ (at 1270 K), for $UC_2$—35 W m$^{-1}$ K$^{-1}$ (at 1270 K), and for $UO_2$—3 W m$^{-1}$ K$^{-1}$ (at 1270 K) as also has been shown in [10].

The properties of $UC_{1-x}N_x$ solid solutions are close to those of individual UN and UC compounds and usually change monotonically depending on the composition. The advanced value of uranium carbonitrides is that they combine some important advantages of both UN and UC. In particular, they were expected to have a lower dissociation pressure than UN and are more resistant to moisture than UC [14,15].

Another important advantage of carbonitrides over individual compounds is the possibility of easier achievement of a single-phase product during industrial production by carbothermal reduction in a stream of nitrogen from $UO_2$.

In addition, the evidence that the introduction of carbon into UN expands the region of homogeneity of compounds was demonstrated in [9]. Taking these features into account, $UC_{1-x}N_x$ solid solutions are considered as independent nuclear fuel along with UN and UC.

Sublimation of $UC_{1-x}N_x$ solid solutions was studied in [4–16]. The authors' results are inconsistent. So, in [9,14], a decrease in the nitrogen pressure over uranium carbonitrides with an increase in the carbon content was found by mass spectrometry, while in [6] no such dependence was found.

In [8], the equilibrium pressures of CO and $N_2$ over the U-C-N-O quaternary system at a temperature of 2000 K were determined. The relevant data are presented in Table 1.

**Table 1.** Composition, cubic lattice period, and equilibrium pressures CO and $N_2$ over the U-C-N-O system at 2000 K [8].

| No | $U_wC_xO_yN_z$ | | | | Cell Parameter, nm | Presure, MPa | |
|---|---|---|---|---|---|---|---|
| | W | x | y | z | | CO | $N_2$ |
| 1 | 0.47 | 0.49 | 0.035 | 0 | 0.4960 | $2.6 \times 10^{-3}$ | 0 |
| 2 | 0.475 | 0.475 | 0.05 | 0 | 0.4960 | $4.6 \times 10^{-3}$ | 0 |
| 3 | 0.48 | 0.34 | 0.047 | 0.13 | 0.49396 | 0.02 | $6.6 \times 10^{-3}$ |
| 4 | 0.485 | 0.36 | 0 | 0.155 | 0.4940 | 0 | 4 |
| 5 | 0.475 | 0.525 | 0 | 0 | 0.4960 | 0 | 0 |
| 6 | 0.475 | 0.505 | 0 | 0.02 | 0.4958 | 0 | $1.3 \times 10^{-4}$ |
| 7 | 0.50 | 0.33 | 0.17 | 0 | 0.4948 | $1.3 \times 10^{-6}$ | 0 |



**Table 1.** *Cont.*

| No | $U_wC_xO_yN_z$ | | | | Cell Parameter, nm | Presure, MPa | |
|---|---|---|---|---|---|---|---|
| 8 | 0.35 | 0.65 | 0 | 0 | - | 0 | 0 |
| 9 | 0.34 | 0.61 | 0.05 | 0 | - | $4.6 \times 10^{-3}$ | 0 |
| 10 | 0.505 | 0 | 0 | 0.495 | 0.4849 | 0 | $5.0 \times 10^{-8}$ |
| 11 | 0.49 | 0 | 0 | 0.51 | 0.4890 | 0 | 0.33 |
| 12 | 0.49 | 0.02 | 0 | 0.49 | 0.48926 | 0 | 1.2 |
| 13 | 0.49 | 0.03 | 0.03 | 0.45 | 0.4894 | 0.2 | 1.2 |
| 14 | 0.49 | 0 | 0.02 | 0.49 | 0.4890 | 0 | 0.3 |
| 15 | 0.50 | 0 | 0.05 | 0.45 | 0.4889 | 0 | $5.3 \times 10^{-8}$ |

One might see that the presence of oxygen in carbonitrides led to the appearance of carbon monoxide in the gas phase. On the other hand, the nitrogen pressure over compositions 4 and 11–14 seems to be too high.

From the data presented, it was impossible to draw unambiguous conclusions about the effect of oxygen impurities in carbonitrides on the dissociation pressure of compounds.

In [17], the thermodynamics of the U-Pu-C-N quaternary system containing oxygen impurities was calculated based on the data available in the literature. Despite a number of assumptions that simplify calculations, for example, the representation of this system in the form of ideal solid solutions, which led to some discrepancy between the experimental and calculated values, it was shown that an increase in the oxygen content in mixed uranium-plutonium carbonitrides led to an increase in the partial pressure of nitrogen at temperatures of 1500 K and 1750 K.

As for UZrCN, it was assumed that this composition has a higher thermal stability than UCN, but experimental data on its behavior at high temperatures are much less. These data was necessary to understand the behavior of the fuel during operation and under conditions of possible emergency overheating.

In [1,2], data were given for the temperature dependences of the partial pressures of uranium and nitrogen over two compositions: $U_{0.9}Zr_{0.1}C_{0.44}N_{0.56}O_{0.01}$ and $U_{0.9}Zr_{0.1}C_{0.62}N_{0.38}O_{0.01}$. For the other compositions, only data on nitrogen pressures have been reported [1]. At the same time, the complete composition of the gas phase has not been determined. In addition, the data on the effect of oxygen impurities on the nature of the sublimation of these compounds were quite insufficient. The appearance of oxygen impurities (0.01–0.5% wt.) was due to the technology for the preparation of these fuels, i.e., carbothermal reduction from uranium and zirconium dioxide in a nitrogen atmosphere [1]. Oxygen in the fuel, depending on the temperature and its content, can be both in the composition of a solid solution of $U_xZr_{x1}C_{x2}N_{x3}O_{x4}$, and also as a separate phase of $UO_2$ [1,4,5,7]. The amount of oxygen impurity in UCN and UZrCN largely determines the characteristics of sublimation, mass transfer, and changes in fuel composition during operation at high temperatures. In [3], the composition of the gas phase was experimentally established by mass spectrometry and the partial pressures over uranium-zirconium carbonitrides with different oxygen contents at a temperature of 2023 K were determined. It is concluded that the pressure in the gas phase of such components as $N_2$, CO, and UO increases with an increase in the oxygen content in the materials. However, the temperature dependences of the partial pressures of all components of the gas phase have not been determined.

The aim of this work was to obtain and analyze experimental data on the sublimation of a number of UZrCN compositions with different contents of carbon, nitrogen, and oxygen impurities.

## 2. Experimental Technique and Materials

The uranium-zirconium carbonitrides taken for research in this work: $U_{0.9}Zr_{0.1}(N_{0.68}C_{0.30}O_{0.015})_{0.92}$, $U_{0.9}Zr_{0.1}(N_{0.48}C_{0.50}O_{0.03})_{0.95}$, $U_{0.9}Zr_{0.1}(N_{0.34}C_{0.63}O_{0.02})_{0.97}$ and $U_{0.1}Zr_{0.9}(N_{0.49}C_{0.50}O_{0.01})_{0.92}$ (correspond to compositions 1–4 in Table 2; listed in lines in Table 3; composition 3 in Table 4; and compositions 13–16, in Table 5), were obtained by the technology of carbothermal reduction from uranium and zirconium dioxide in a nitrogen atmosphere, as was described earlier [1].

**Table 2.** The chemical composition, cubic lattice period of the investigated uranium-zirconium carbonitrides.

| No | Formula Composition | Cell Parameter, nm | Content of the Elements, Mass. % | | | | |
|----|---------------------|--------------------|------|------|------|------|------|
| | | | U | Zr | N | C | O |
| 1 | $U_{0.9}Zr_{0.1}(N_{0.68}C_{0.30}O_{0.015})_{0.92}$ | 0.4898 | 90.0 | 4.0 | 3.7 | 1.4 | 0.09 |
| 2 | $U_{0.9}Zr_{0.1}(N_{0.48}C_{0.50}O_{0.03})_{0.95}$ | 0.4928 | 91.0 | 3.7 | 2.7 | 2.4 | 0.17 |
| 3 | $U_{0.9}Zr_{0.1}(N_{0.34}C_{0.63}O_{0.02})_{0.97}$ | 0.4938 | 91.0 | 3.8 | 2.0 | 3.1 | 0.15 |
| 4 | $U_{0.1}Zr_{0.9}(N_{0.49}C_{0.50}O_{0.01})_{0.92}$ | 0.4926 | 19.5 | 69.1 | 5.5 | 4.6 | 0.1 |

**Table 3.** Parameters of the temperature dependence of the partial pressures of the gas phase components over uranium-zirconium carbonitrides in the form $lgP(M\Pi a) = A - B/T$ and the sublimation heat (this work).

| Molecule | Temperature Range, K | $A \pm \Delta A$ | $B \pm \Delta B$ | $\Delta H_{S,T}$ kJ/mol |
|----------|----------------------|------------------|------------------|--------------------------|
| $U_{0.9}Zr_{0.1}(N_{0.68}C_{0.30}O_{0.015})_{0.92}$ (sample No 1, Table 1 and sample No 13, Table 5) | | | | |
| U | 1773–2323 | $6.4 \pm 0.2$ | $29{,}200 \pm 400$ | $560 \pm 8$ |
| UO | 1813–2323 | $6.4 \pm 0.3$ | $30{,}600 \pm 600$ | $590 \pm 10$ |
| $UO_2$ | 1913–2323 | $2.2 \pm 0.3$ | $23{,}700 \pm 900$ | $450 \pm 20$ |
| UN | 1973–2323 | $5.6 \pm 0.9$ | $34{,}600 \pm 2000$ | $660 \pm 40$ |
| $N_2$ | 1863–2323 | $11.1 \pm 0.4$ | $35{,}100 \pm 900$ | $670 \pm 20$ |
| $U_{0.9}Zr_{0.1}(N_{0.48}C_{0.50}O_{0.03})_{0.95}$ (sample No 2, Table 1 and sample No 14, Table 5) | | | | |
| U | 1773–2223 | $6.6 \pm 0.2$ | $29{,}500 \pm 500$ | $565 \pm 8$ |
| UO | 1813–2223 | $6.1 \pm 0.4$ | $30{,}400 \pm 1000$ | $580 \pm 20$ |
| $UO_2$ | 1913–2223 | $5.3 \pm 0.6$ | $30{,}000 \pm 2000$ | $570 \pm 40$ |
| Zr | 2123–2223 | $9.1 \pm 1.5$ | $41{,}000 \pm 3500$ | $790 \pm 70$ |
| $N_2$ | 1873–2223 | $11.2 \pm 0.5$ | $35{,}600 \pm 1000$ | $680 \pm 20$ |
| $U_{0.9}Zr_{0.1}(N_{0.34}C_{0.63}O_{0.02})_{0.97}$ (sample No 3, Table 1 and sample No 15, Table 5) | | | | |
| U | 1773–2323 | $6.7 \pm 0.2$ | $30{,}300 \pm 400$ | $580 \pm 8$ |
| UO | 1813–2323 | $7.0 \pm 0.3$ | $32{,}900 \pm 600$ | $630 \pm 10$ |
| $UO_2$ | 1913–2323 | $2.0 \pm 0.5$ | $23{,}400 \pm 1000$ | $440 \pm 20$ |
| UN | 1973–2323 | $7.0 \pm 0.8$ | $38{,}100 \pm 1500$ | $730 \pm 30$ |
| $N_2$ | 1863–2323 | $11.5 \pm 0.4$ | $36{,}300 \pm 900$ | $700 \pm 20$ |

**Table 3.** *Cont.*

| U$_{0.1}$Zr$_{0.9}$(N$_{0.49}$C$_{0.50}$O$_{0.01}$)$_{0.92}$ (sample No 4, Table 1 and sample No 16, Table 5) | | | |
|---|---|---|---|
| U | 1773–2323 | 6.1 ± 0.1 | 28,600 ± 200 | 550 ± 5 |
| UO | 1813–2323 | 4.0 ± 0.5 | 26,700 ± 1000 | 510 ± 20 |
| UO$_2$ | 1913–2323 | 2.5 ± 1.0 | 25,500 ± 2200 | 490 ± 40 |
| UN | 1973–2323 | 11.0 ± 2.0 | 47,000 ± 5000 | 900 ± 100 |
| Zr | 2123–2323 | 10.8 ± 1.2 | 44,000 ± 3000 | 840 ± 50 |
| N$_2$ | 1863–2323 | 9.7 ± 0.4 | 32,400 ± 800 | 620 ± 20 |

**Table 4.** Parameters of the temperature dependence of the partial pressures of uranium and nitrogen in the form $\lg P$(mm Hg) $= A - B/T$ according to [1].

| | U$_{0.9}$Zr$_{0.1}$(C$_{0.44}$N$_{0.56}$O$_{0.01}$)$_{0.71}$ | | |
|---|---|---|---|
| **Molecule** | $A \pm \Delta A$ | $B \pm \Delta B$ | $\Delta H_{S,T}$ kJ/mol |
| U | 8.1 ± 0.03 | 23,500 ± 500 | 458 |
| N$_2$ | 11.6 ± 0.03 | 30,000 ± 500 | 573 |
| | U$_{0.9}$Zr$_{0.1}$(C$_{0.62}$N$_{0.38}$O$_{0.01}$)$_{0.97}$ | | |
| U | 11.22 ± 0.03 | 32,300 ± 500 | 619 |
| N$_2$ | 11.9 ± 0.03 | 29,600 ± 500 | 564 |

**Table 5.** Composition and equilibrium pressures of components over UCN and UZrCN with different contents of carbon, nitrogen, and oxygen impurities at high temperatures according to different authors.

| No. | Composition | T, K | Presure, MPa | | | | | | | Ref. |
|---|---|---|---|---|---|---|---|---|---|---|
| | | | N$_2$ | U | UO | UO$_2$ | UN | Zr | CO | |
| 1 | U(N$_{0.823}$C$_{0.14}$O$_{0.037}$)$_{1.00}$ | 2173 | - | | | | | | $1 \times 10^{-6}$ | 5 |
| 2 | U(N$_{0.666}$C$_{0.32}$O$_{0.014}$)$_{1.00}$ | 2173 | | | | | | | $6 \times 10^{-7}$ | 5 |
| 3 | U(N$_{0.564}$C$_{0.43}$O$_{0.006}$)$_{1.00}$ | 2173 | | | | | | | $5.5 \times 10^{-7}$ | 5 |
| 4 | UN$_{0.30}$C$_{0.7}$ | 2000 | $5.6 \times 10^{-9}$ | $4.9 \times 10^{-10}$ | | | | | 0 | 9 |
| 5 | UN$_{0.48}$C$_{0.52}$ | 2000 | $2.0 \times 10^{-8}$ | $1.5 \times 10^{-9}$ | | | | | 0 | 9 |
| 6 | U(N$_{0.52}$C$_{0.45}$O$_{0.03}$)$_{0.98}$ | 2000 | $3.6 \times 10^{-6}$ | $8.9 \times 10^{-9}$ | $7.9 \times 10^{-9}$ | $5.0 \times 10^{-10}$ | $5.6 \times 10^{-11}$ | - | | 4 |
| 7 | U(N$_{0.27}$C$_{0.69}$O$_{0.05}$)$_{1.01}$ | 2000 | $4.5 \times 10^{-6}$ | $7.9 \times 10^{-9}$ | $5.6 \times 10^{-9}$ | $2.2 \times 10^{-10}$ | $4.0 \times 10^{-11}$ | - | | 4 |
| 8 | U$_{0.9}$Zr$_{0.1}$(C$_{0.44}$N$_{0.56}$O$_{0.01}$)$_{0.71}$ | 2000 | $5.8 \times 10^{-8}$ | $2.9 \times 10^{-8}$ | | | | | | 1 |
| 9 | U$_{0.9}$Zr$_{0.1}$(C$_{0.62}$N$_{0.38}$O$_{0.01}$)$_{0.97}$ | 2000 | $1.7 \times 10^{-7}$ | $1.6 \times 10^{-9}$ | | | | | | 1 |
| 10 | U$_{0.9}$Zr$_{0.1}$(N$_{0.596}$C$_{0.397}$O$_{0.008}$)$_{0.917}$ | 2023 | $1.2 \times 10^{-7}$ | $6.6 \times 10^{-9}$ | $5.5 \times 10^{-10}$ | | | $1.1 \times 10^{-8}$ | | 3 |
| 11 | U$_{0.91}$Zr$_{0.09}$(N$_{0.475}$C$_{0.475}$O$_{0.051}$)$_{0.99}$ | 2023 | $2.2 \times 10^{-6}$ | $5.7 \times 10^{-9}$ | $1.9 \times 10^{-9}$ | | | $3.3 \times 10^{-7}$ | | 3 |
| 12 | U$_{0.9}$Zr$_{0.1}$(N$_{0.476}$C$_{0.422}$O$_{0.102}$)$_{0.98}$ | 2023 | $2.7 \times 10^{-6}$ | $7.4 \times 10^{-9}$ | $4.9 \times 10^{-9}$ | | | $4.6 \times 10^{-7}$ | | 3 |
| 13 | U$_{0.9}$Zr$_{0.1}$(N$_{0.68}$C$_{0.30}$O$_{0.01}$)$_{0.92}$ | 2000 | $3.6 \times 10^{-7}$ | $6.3 \times 10^{-9}$ | $1.3 \times 10^{-9}$ | $2.2 \times 10^{-10}$ | $2.0 \times 10^{-12}$ | Trace | - | This work |
| 14 | U$_{0.9}$Zr$_{0.1}$(N$_{0.48}$C$_{0.50}$O$_{0.03}$)$_{0.95}$ | 2000 | $2.5 \times 10^{-7}$ | $7.1 \times 10^{-9}$ | $8.0 \times 10^{-10}$ | $2.0 \times 10^{-10}$ | Trace | $4.0 \times 10^{-12}$ | - | This work |
| 15 | U$_{0.9}$Zr$_{0.1}$(N$_{0.34}$C$_{0.63}$O$_{0.02}$)$_{0.97}$ | 2000 | $2.2 \times 10^{-7}$ | $3.6 \times 10^{-9}$ | $3.5 \times 10^{-10}$ | $2.0 \times 10^{-10}$ | $8.9 \times 10^{-13}$ | Trace | - | This work |
| 16 | U$_{0.1}$Zr$_{0.9}$(N$_{0.49}$C$_{0.50}$O$_{0.01}$)$_{0.92}$ | 2000 | $3.2 \times 10^{-7}$ | $6.3 \times 10^{-9}$ | $4.5 \times 10^{-10}$ | $5.6 \times 10^{-11}$ | $3.2 \times 10^{-13}$ | $6.3 \times 10^{-12}$ | This work |

Oxygen impurities in glove-box gas phase were within the range 2–4 ppm O$_2$ as controlled with GPR-1200 Portable O$_2$ PPM Analyzer ATEX version. The materials were grains with a particle size of 0.1–0.5 mm. According to X-ray analysis, the presence of uranium dioxide in the initial state was not detected. Table 2 presents the chemical

composition of the materials under study according to the chemical analyses. For each composition, 3 experimental tests were undertaken.

Sublimation was studied on an MS-1301 mass spectrometer. A portion of the studied material (500–1133 mg) was placed in "Knudsen cells" made of polycrystalline tungsten, with a ratio of sublimation and effusion areas equal to 400, which provided a quasi-equilibrium mode of sublimation. The cells were heated by electron bombardment. The temperature was measured using a calibrated micropyrometer OMP-054.

Each experimental sample was tested in 3–4 runs of cyclic rise of the temperature to the maximum test value and descent to the room temperature. The most stable values recorded in the 3 or 4 cycles were used for calculations and presented in Figures 1–4.

The magnetic mass analyzer applied for the bulk of the compositions had a general resolution of about 600.

The samples with the compositions 10–12 (Table 5) were studied by high-resolution mass spectrometry (the resolution of the quadrupole mass analyzer was ~4000), which made it possible to separate carbon monoxide and nitrogen in the mass spectrum (MX-1320 mass spectrometer) [3]. The ion current was recorded with an electrometric amplifier and an electron multiplier. The residual pressure in the device was no worse than $4.3 \times 10^{-5}$ Pa at the maximum temperature of the experiment. For high-resolution mass spectrometry, the corresponding pressure was no worse than $1.3 \times 10^{-6}$ Pa [3].

The absolute values of partial pressures were determined by calibrating the instrument against silver (107 amu). A weighed portion (30–50 mg) of the calibration material (99.9999 wt% Ag) was placed directly into the cell together with the test material. The instrument was calibrated at the beginning of each experiment. All materials were loaded in a sealed box in an environment of dried high-purity argon.

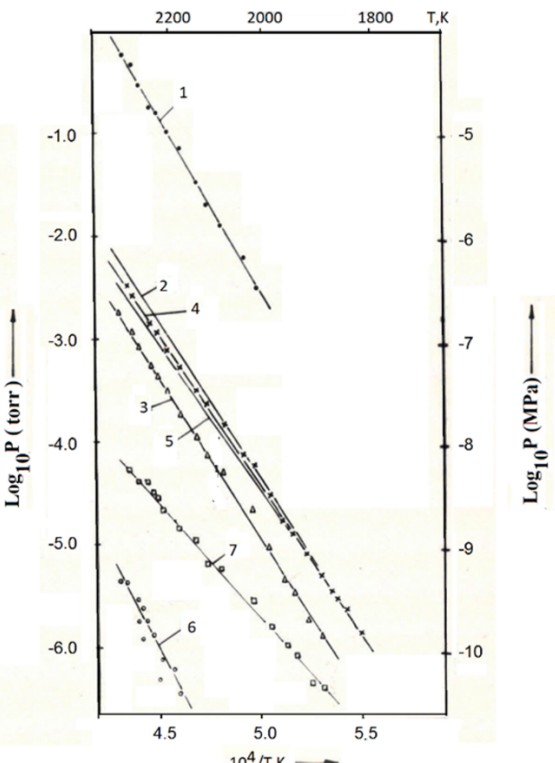

**Figure 1.** Dependence of partial pressures of gas phase components on temperature in Arrhenius coordinates over composition of sample 1 in Table 2 or No. 13 in Table 5: 1—nitrogen, 2—total pressure U + UO, 3—uranium monoxide, 4—uranium, 5—uranium from uranium nitride, 6—uranium mononitride, 7—uranium dioxide.

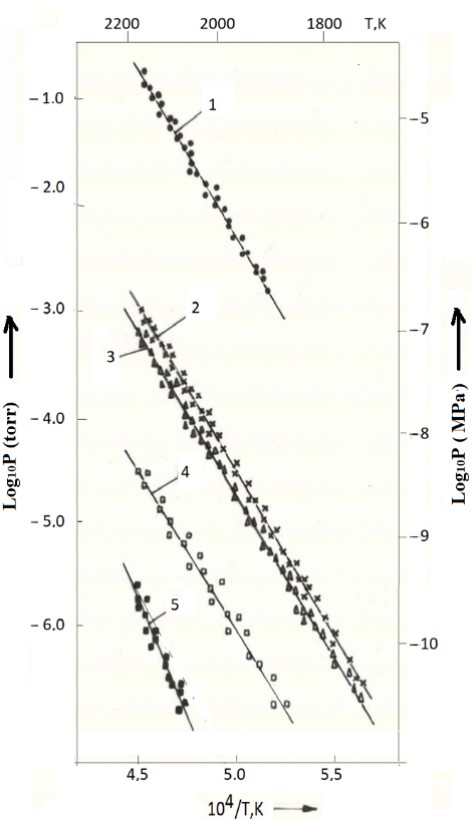

**Figure 2.** Dependence of partial pressures of gas phase components on temperature in Arrhenius coordinates over composition of sample 2 in Table 2 or No. 14 in Table 5: 1—nitrogen, 2—uranium, 3—uranium monoxide, 4—uranium dioxide, 5—zirconium.

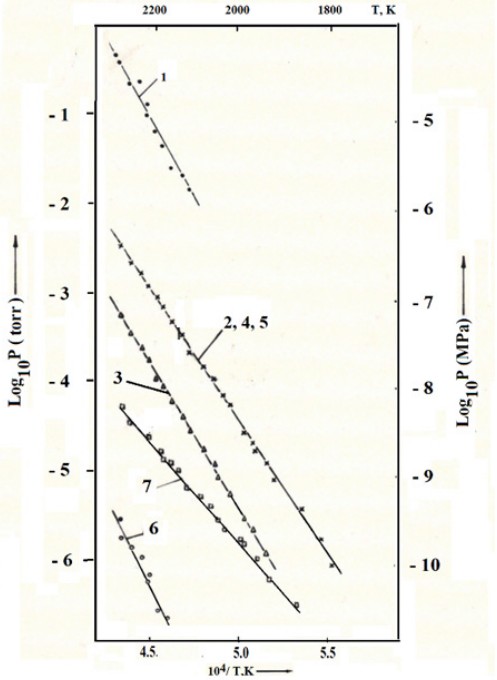

**Figure 3.** Dependence of partial pressures of gas phase components on temperature in Arrhenius coordinates over composition of sample 3 in Table 2 or No. 15 in Table 5: 1—nitrogen, 2,4,5—uranium, 3—uranium monoxide, 6—uranium mononitride, 7—uranium dioxide.

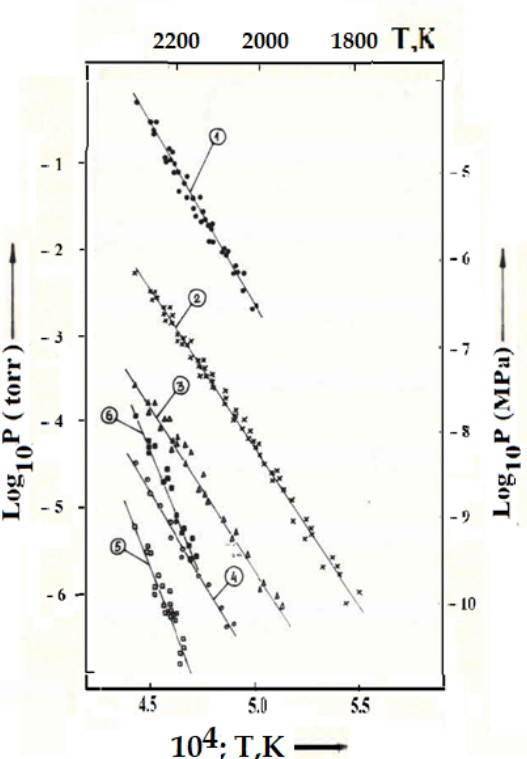

**Figure 4.** Dependence of partial pressures of gas phase components on temperature in Arrhenius coordinates over composition of sample 4 in Table 2 or No. 16 in Table 5: 1—nitrogen, 2—uranium, 3—uranium monoxide, 4—uranium dioxide, 5—uranium mononitride, 6—zirconium.

The partial pressures of the gas phase components were calculated using the formula:

$$P_i = P_{Ag} \times \sigma_{Ag}\beta_{Ag}\gamma_{Ag} \times (\sigma_i\beta_i\gamma_i)^{-1} \times I_iT_i \times (I_{Ag}T_{Ag})^{-1}$$

where *i*—the index of the corresponding component of the gas phase, $P_i$ was the partial pressure of the *i*-th component, $I_i$ was the ion current of the *i*-th component, *T* was the temperature, $\sigma_i$ was the ionization cross section of molecules of the *i*-th component, $\beta_i$ was the multiplication factor for molecules of the *i*-th component, $\gamma_i$—natural abundance of the *i*-th component. Similar values with the index $A_g$ referred to the calibration substance—silver. For the $P_{Ag}$ values we used the data from [18], for σi(Ag)—from [19], and for γi(Ag)—from [20].

Since the partial pressure of uranium monoxide in some experiments was comparable in magnitude with the pressure of uranium, the obtained $P_U$ values were corrected for the dissociation of UO molecules under the action of an ionizing electron beam. It was experimentally established that the contribution to the uranium ion current from the dissociative ionization of UO molecules is $I_{U+} = (0.4–0.5)I_{UO+}$, where *I*—the value of the measured ion current.

The temperature dependences of the partial pressures of the gas phase components (in Arrhenius coordinates), as well as the errors of the corresponding quantities, were calculated using the least-squares method.

### 3. Research Results

The experimental dependences of the partial pressures of the gas phase components on the reciprocal temperature over the studied compositions are shown in Figures 1–4. The data were obtained during the rise to or descent from the maximum temperature of the experiment.

Figure 1 shows the experimental dependences of the partial pressures of the gas phase components on the reciprocal temperature over one of the samples of composition, No. 1 (No. 13 in Table 5). The data were obtained during the descent from the maximum temperature of the experiment. A comparison of the obtained results with the data on other samples showed that the difference between them is insignificant and is included in the measurement error. The study of sublimation of uranium-zirconium carbonitride was carried out in the temperature range 1773–2323 K. The initial sample weight was 638 mg, and the residence time of the fuel at temperatures above 1700 K was about 10.5 h. The decrease in the mass of the sample during the experiment was 3.7 mg.

The main components of the gas phase were (in descending order of partial pressures): nitrogen, uranium, uranium monoxide, uranium dioxide and uranium mononitride (Figure 1). Traces of atomic zirconium were also observed at the sensitivity limit of the device, but it was not possible to obtain the temperature dependence of its partial pressure.

In addition to the listed components of the gas phase, Figure 1 shows the total pressure of uranium-containing components (U + UO) and the pressure of uranium from uranium-zirconium carbonitride, which was obtained by subtracting the contribution from dissociative ionization by an electron beam of UO molecules ($I_{U+} = (0.4–0.5)I_{UO+}$), from the detected uranium ion current.

Figure 2 shows the experimental dependences of the partial pressures of the gas phase components on the reciprocal temperature over composition No. 2 (No. 14 in Table 5). The data were obtained both on the ascent and during the descent from the maximum temperature of the experiment in the temperature range of 1773–2223 K. The initial sample weight was 626 mg, and the residence time of the fuel at temperatures above 1700 K was about 10 h. The decrease in the mass of the sample during the experiment was 3.5 mg. The main gas phase components were (in descending order of partial pressures): nitrogen, uranium, uranium monoxide, uranium dioxide and zirconium (Figure 2). Traces of uranium mononitride were also observed, but it was not possible to obtain the temperature dependence of its partial pressure.

The study of sublimation of uranium-zirconium carbonitride composition No. 3 (No. 15, Table 5) was carried out in the temperature range of 1773–2323 K. The initial sample weight was 525 mg, and the residence time of the fuel at temperatures above 1700 K was about 18 h. The decrease in the sample mass during the experiment was 4.5 mg. The components of the gas phase were (in descending order of partial pressures): nitrogen, uranium, uranium monoxide, uranium dioxide, uranium mononitride (Figure 3). Traces of atomic zirconium were also observed. The corresponding experimental data obtained during the descent from the maximum temperature of the experiment are shown in Figure 3.

The study of sublimation of uranium-zirconium carbonitride composition No. 4 (No. 16, Table 5) was carried out in the temperature range of 1773–2323 K. The initial sample weight was 1133 mg, and the residence time of the fuel at temperatures above 1700 K was about 11 h. The decrease in the mass of the sample during the experiment was 4.0 mg. The corresponding experimental data obtained both on the ascent and on the descent from the maximum temperature of the experiment are shown in Figure 4. The gas phase components were (in descending order of partial pressures) as follows: nitrogen, uranium, uranium monoxide, atomic zirconium, uranium dioxide, uranium mononitride (Figure 4).

Table 3 lists the parameters of the semi-logarithmic dependences "pressure-temperature" of the components of the gas phase over the four studied compositions calculated by the least-squares method, as well as the corresponding heats of sublimation and the root-mean-square measurement errors.

Table 4 shows the data [1] on the pressures of uranium and nitrogen over UZrCN. Table 5 shows the composition and equilibrium pressures of components over UCN and UZrCN with different contents of carbon, nitrogen, and oxygen impurities at high temperatures according to data from different authors, including the results of this work.

The obtained data should be compared with those presented in the literature. In [1], the partial pressures of uranium and nitrogen were simultaneously measured over uranium-

zirconium carbonitrides of two compositions. The results thus obtained are presented in Table 3.

Despite certain discrepancies in the values of the parameters of temperature dependences, especially for nitrogen, the absolute values of pressures at a temperature of 2000 K for both uranium and nitrogen turned out to be close to our data (Table 5).

## 4. Discussion

The appearance of the detected components in the gas phase can be caused by the passage of the following chemical reactions in uranium-zirconium carbonitrides containing oxygen at high temperatures (above 1800 K):

$$UZrC_{1-x}N_{x(sld)} \rightarrow x/2U_{(l \cdot g)} + x/2UN_{(g)} + x/4N_{2(g)} + xZr_{(g)} + (1-x)UZrC_{(sld)} \quad (1)$$

$$UZrN_{1-y}O_{y(sld)} \rightarrow (1-y)U_{(l \cdot g)} + yUO_{(g)} + (1-y)/2N_{2(g)} + Zr_{(g)} \quad (2)$$

$$UZrC_{1-z}O_{z(sld)} \rightarrow 2zU_{(l \cdot g)} + zCO_{(g)} + 2z\ Zr_{(g)} + (1-2z)UZrC_{(sld)} \quad (3)$$

$$UZrC_{1-w}O_{w(sld)} \rightarrow w/2U_{(l \cdot g)} + w/2UO_{2(g)} + wZr_{(g)} + (1-w)UZrC_{(sld)} \quad (4)$$

The above scheme of sublimation reactions was fully confirmed by mass spectrometry data and, despite its idealized nature (for example, the change in stoichiometry during sublimation, etc., was not taken into account), in general, reflected the following features of the process:

1.  Sublimation of uranium-zirconium carbonitrides occurs incongruently with a predominant loss of nitrogen, which led to a shift in the initial composition towards a richer carbon;
2.  Oxygen impurities contribute to the additional release of nitrogen and the appearance of oxide components UO, $UO_2$ and CO in the gas phase;
3.  In the gas phase above uranium-zirconium carbonitrides, in addition to $N_2$, U, UO, $UO_2$ and UN, one more component is found—atomic zirconium.

The first two features of the sublimation of uranium-zirconium carbonitrides mentioned above were similar to those established for UCN [4].

In the gas phase above the studied materials, zirconium dioxide was not recorded. Zirconium monoxide was probably reduced by carbonitride to zirconium, which led to the absence of ZrO(g) in the mass spectrum. This caused the appearance of atomic zirconium in the mass spectrum. Reliable registration of its ion current was possible only for compositions No. 14 and No. 16 (Table 5). For compositions No. 13 and No. 15 (Table 5), traces of atomic zirconium were detected at the maximum temperatures of the experiments at the sensitivity limit of the device. The enthalpy of sublimation of zirconium for compositions No. 14 and No. 16 turned out to be very high, at $(790 \pm 70)$ and $(840 \pm 50)$ kJ/mol, which indicates a greater difficulty in passing reactions (1–4), compared with reactions (3–6) [4].

Comparison of data on the sublimation of uranium carbonitrides (Table 5, compositions 6 and 7) and uranium-zirconium carbonitrides (Table 5, compositions 13–15) showed that the introduction of zirconium carbonitrides and an increase in the carbon content in them led to a decrease in the pressure of uranium and nitrogen monoxide (Figure 5). This phenomenon can be explained by the fact that zirconium in uranium carbonitride binds oxygen impurities into a thermally stable zirconium oxycarbonitride with a wider (in terms of oxygen) homogeneity region in comparison with uranium carbonitride that does not contain zirconium.

Comparison of the values of partial pressures of components over uranium-zirconium carbonitrides (Table 5, compositions 10–12) showed that an increase in the oxygen content in them led to an increase in pressures in the gas phase of such components as $N_2$, CO, UO [3].

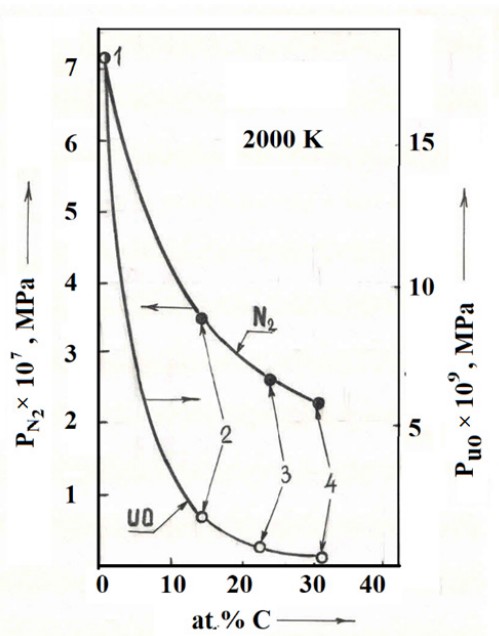

**Figure 5.** Change in the partial pressures of N$_2$ and UO over uranium-zirconium carbonitrides depending on the carbon content at a temperature of 2000 K, in comparison with the data on uranium mononitride [21]. 1—UN [21], 2—composition No. 13 (this work), 3—composition No. 14 (this work), 4—composition No. 15 (this work) (Table 5).

The appearance of carbon monoxide in the gas phase led to spectral noise in determining the partial pressure of nitrogen. However, in [3], it was shown by high-resolution mass spectrometry (separation of CO and N$_2$ molecules in the mass spectrum) that at an oxygen content of 0.04–0.5% at., the contribution of carbon monoxide to the nitrogen pressure over uranium-zirconium carbonitrides at a temperature of 2023 K did not exceed 15% and decreased with exposure. The presented here results referred to the third or fourth temperature cycles (rise to the maximum temperature of the experiment, followed by a stepwise descent). Therefore, it can be concluded that the contribution of carbon monoxide to the partial pressure of nitrogen in this work never exceeded 10–15%.

The results obtained in this work may be valuable for the prediction of the fission product reactions and speciation during the irradiation of this nuclear fuel in the nuclear reactor if we include the thermodynamic properties and time dependences of this FP based on the data presented in [22].

## 5. Conclusions

An analysis of the experimental data obtained in this work and their comparison with the literature data on the sublimation of uranium-zirconium carbonitrides with different contents of carbon, nitrogen, and oxygen impurities at high temperatures showed that the sublimation of these materials occurs incongruently with a predominant loss of nitrogen, which led to a shift in their initial compositions in the direction of the phase that was richer in carbon. The gas phase in this case had a complex composition, which included: nitrogen, uranium, uranium monoxide, uranium dioxide, uranium mononitride and atomic zirconium. The considered chemical mechanism of sublimation of the studied materials indicated that oxygen impurities in them contributed to the additional release of nitrogen and the appearance of oxide components in the gas phase: UO, UO$_2$ and CO. The introduction of zirconium carbonitrides and an increase in the carbon content (within the concentration regions $U_{(0.9-1.0)}Zr_{(0-0.1)}(C_{(0.3-0.63)}N_{(0.34-0.68)}O_{(0.01-0.03)})_{(0.92-0.97)})$ contributed to a decrease in the partial pressures of uranium monoxide and nitrogen, thereby increasing the thermal stability of this kind of nuclear fuel. So, this nuclear fuel has prospective applications in reactors with high burn-up rates.

**Author Contributions:** Conceptualization, G.S.B. and K.E.G.; methodology, G.S.B. and K.E.G.; software, G.S.B.; validation, G.S.B.; formal analysis, G.S.B.; investigation, G.S.B.; resources, K.E.G.; data curation, G.S.B.; writing—original draft preparation, G.S.B.; writing—review and editing, K.E.G.; visualization, K.E.G.; supervision, K.E.G.; project administration, K.E.G.; funding acquisition, K.E.G. All authors have read and agreed to the published version of the manuscript.

**Funding:** X-ray diffraction experiments were performed at the Center for Shared Use of Physical Methods of Investigation at the Frumkin Institute of Physical Chemistry and Electrochemistry, RAS. The study was supported by the Ministry of Science and Higher Education of the Russian Federation (program no. 122011300061-3).

**Data Availability Statement:** The data presented in this study are available on request from the corresponding author.

**Conflicts of Interest:** The authors declare no conflict of interest.

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
