# Peer review of "New Experimental Data on Partial Pressures of Gas Phase Components over Uranium-Zirconium Carbonitrides at High Temperatures and Its Comparative Analysis"

_jne, doi:10.3390/jne3040022_

Round 1
Reviewer 1 Report
The article is of interest to the nuclear engineering and materials community as it establishes the sublimation mechanisms of UCN and UZrCN fuels. Some major corrections are required before publication.
Overall - the English (grammar and phrasing) needs to be improved and in at least 3 instances there are Russian symbols/words in the text. Please review carefully and remove/translate to English.
Introduction:
This needs to be improved with 1) more background of why the UCN and UZrCN materials are of interest to the reader (one sentence is not sufficient) and 2) more background of the previous work performed on the topic and the reported mechanisms at play that dictate that oxygen impurities have an effect on the thermal stability of UCN. There is one statement "The amount of oxygen impurity in UCN...operation at high temperature" which needs more explanation and more references put to it. Also, what amount of oxygen impurities? Further explanation is needed.
Experimental methods:
There is very little mention of how the compounds that were investigated were made and characterized. More details on the carbothermic reduction is required and more details on how the particle sizes and XRD was obtained. Please include at a minimum a figure of the obtained XRD spectrums for the compositions you synthesized. An image of the obtained grains would also be a nice addition.
One very important point is that the stoichiometry of the C, N, or O was not established. I understand that you synthesized it as a specific composition but this was never proven by any measurement that was reported here. How do you know that the C, O, or N amount is what you think it is? This has fairly large repercussions for the results. Please state clearly if any measurements were done and if not provide an explanation as to why you believe that the achieved composition is exactly what was targeted.
Also, you have stated that high purity argon was used for the sublimation experiments. It seems that controlling the levels of oxygen impurities in the stream is very important in these experiments. Did you measure the parts per million amount of O in the high-purity Argon stream? Was the gas atmosphere further purified to ensure a very low level of ppm O in the stream?
Discussion:
At the end of the discussion section you attempt to cite a journal but instead it read "[]". Please correct.
Are figures 1-5 all based on literature data? It isn't clear what is new data and what isn't. Please clarify in the text and in the figure captions.
Conclusion:
Would appreciate if in the conclusion ranges for oxygen, zirconium and carbon additions were given. For example you say that incorporation of Zr and additions of C lead to a more thermally stable fuel - what are the compositional ranges for this? How much Zr and C has to be added?
A broader perspective on the use and development of these fuels would be appreciated.
Author Response
Please see details in the attachment.

Reviewer 2 Report
The manuscript is written in present tense it should be written in past or past perfect tense.
The introduction and structure are weak, only a few references are cited. It does not explain the theme of the work properly.
Explanation of Experimental Technique is also weak with a little presentation of methodology.
Results and discussion need more explanation about the results and outcomes with their reasoning. In the discussion section, the author does not discuss Figures 1-4 in detail. High-resolution figures are required (300dpi).
There is no error evaluation in the experimental measurement. If there is error evaluation the methodology (error propagation) needed to be explained. In one curve of the graph (Fig. 3) error bar is shown, which is quite a bib error. It needs to be explained.
The heading should be properly numbered.
Author Response
Please see details in the attachment.

Round 2
Reviewer 1 Report
The introduction has been updated however it still requires work on grammar and format (paragraph style is inconsistent) - please note that it is not "leaded to" but "led to" for example. This has been used incorrectly throughout the manuscript (including abstract, introduction, and conclusions).
Some Russian symbols still remain - please update carefully.
Table 1 - What is the unit for the Cell Parameter? Please update the table and use "." instead of "," for the decimal points.
Experimental technique:
You reference [1] (Alexeev, Zaitsev, 2013) however this is not available online. Does this publication include the XRD analysis?
You mention X-ray analysis without presenting that data - you cannot make this statement unless this data is provided. Your response that this data cannot be inlcuded is not satisfactory. Please provide an explanation. If you absolutely cannot give details of the preparation technique then you must provide data on the characterization of the materials. XRD is necessary.
Characterization of the oxygen impurities in your analyzed materials is required to support your arguments. How was this characterized? This must be discussed. This was not addressed in your response.
From your response "
As indicated in the text - all materials were loaded in a sealed box in an environment of dried high-purity argon. The sublimation experiments were conducted in high vacuum as needed imperatively for the Knudsen camera tests. (Peter Atkins and Julio de Paula, Physical Chemistry (8th ed., W.H.Freeman 2006) p.756 ISBN 0-7167-8759-8) |
" - You have not addressed my question. What was the O2 environment in the glovebox? Specifically ppm of O2. I understand a sealed box with a dried high-purity argon atmosphere was used. This can range - what specifically was your atmosphere.
Figure 5 - my comment was not addressed. Please re-work this figure to clearly indicate which data is literature data and which is new data.
Author Response
Answers to the reviewer 1
- The book referenced as 1 ) Alexeev S.V., Zaitsev V.A. Nitride fuel for nuclear energy: M.: Technosphere. 2013. 240 p. is now easily available from the publisher at https://urss.ru/cgi-bin/db.pl?lang=Ru&blang=ru&page=Book&id=182646 (the book is written in Russian). Done.
- All mistakes with the verb lead in past form, stile of the characters and Russian characters are corrected in the text. Done.
- Reviewer ask that we must provide data on the characterization of the materials. WE added the chemical composition data (in table 2). XRD may not be presented. Production and characterization of the material provided for this study was made by the authors of ref. 1. Our task was to carry out sublimation tests. Done in possible part.
- The O2 concentrations is specified in Experimental. Done
- Table 1 : Cell parameter, is given in nm Done
- Fig 5 – legend is detailed. Done